# Molecular Insight into *Mycobacterium tuberculosis* Resistance to Nitrofuranyl Amides Gained through Metagenomics-like Analysis of Spontaneous Mutants

**DOI:** 10.3390/ph15091136

**Published:** 2022-09-12

**Authors:** Igor Mokrousov, Ivaylo Slavchev, Natalia Solovieva, Marine Dogonadze, Anna Vyazovaya, Violeta Valcheva, Aleksey Masharsky, Olesya Belopolskaya, Simeon Dimitrov, Viacheslav Zhuravlev, Isabel Portugal, João Perdigão, Georgi M. Dobrikov

**Affiliations:** 1Laboratory of Molecular Epidemiology and Evolutionary Genetics, St. Petersburg Pasteur Institute, 197101 St. Petersburg, Russia; 2Henan International Joint Laboratory of Children’s Infectious Diseases, Children’s Hospital Affiliated to Zhengzhou University, Zhengzhou 450018, China; 3Institute of Organic Chemistry with Centre of Phytochemistry, Acad. G. Bonchev Street, bl. 9, 1113 Sofia, Bulgaria; 4St. Petersburg Research Institute of Phthisiopulmonology, 191036 St. Petersburg, Russia; 5The Stephan Angeloff Institute of Microbiology, Bulgarian Academy of Sciences, Acad. G. Bonchev Street, bl. 26, 1113 Sofia, Bulgaria; 6Resource Center “Bio-bank Center”, Research Park of St. Petersburg State University, 198504 St. Petersburg, Russia; 7iMed.ULisboa–Instituto de Investigação do Medicamento, Faculdade de Farmácia, Universidade de Lisboa, 1649004 Lisbon, Portugal

**Keywords:** nitrofuranyl amides, *mycobacterium tuberculosis*, spontaneous mutagenesis, whole-genome sequencing

## Abstract

We performed synthesis of new nitrofuranyl amides and investigated their anti-TB activity and primary genetic response of mycobacteria through whole-genome sequencing (WGS) of spontaneous resistant mutants. The in vitro activity was assessed on reference strain *Mycobacterium tuberculosis* H37Rv. The most active compound **11** was used for in vitro selection of spontaneous resistant mutants. The same mutations in six genes were detected in bacterial cultures grown under increased concentrations of **11** (2×, 4×, 8× MIC). The mutant positions were presented as mixed wild type and mutant alleles while increasing the concentration of the compound led to the semi-proportional and significant increase in mutant alleles. The identified genes belong to different categories and pathways. Some of them were previously reported as mediating drug resistance or drug tolerance, and counteracting oxidative and nitrosative stress, in particular: *Rv0224c*, *fbiC*, *iniA*, and *Rv1592c*. Gene-set interaction analysis revealed a certain weak interaction for gene pairs *Rv1592–Rv1639c* and *Rv1592–Rv0224c*. To conclude, this study experimentally demonstrated a multifaceted primary genetic response of *M. tuberculosis* to the action of nitrofurans. All three **11**-treated subcultures independently presented the same six SNPs, which suggests their non-random occurrence and likely causative relationship between compound action and possible resistance mechanism.

## 1. Introduction

Tuberculosis (TB), caused by *Mycobacterium tuberculosis*, is both an ancient and reemerging human disease. The estimated numbers of new TB cases and TB deaths were 10 and 1.5 million, respectively, in 2020 [1]. Despite sufficiently effective treatment regimens, as well as prevention and control TB measures, there is an active transmission and high prevalence of resistant *M. tuberculosis* strains. This situation continues to be a major public health problem in both developed and developing countries, particularly aggravated by the emergence and spread of multidrug-resistant (MDR) and extensively drug-resistant (XDR) *M. tuberculosis* strains.

*M. tuberculosis* has a complex and chemically resistant cell wall. Therefore, anti-tubercular drugs are specific and do not act on other pathogenic bacteria, and vice versa—many available antibiotics do not (with a few exceptions) act on mycobacteria. Many anti-TB antibiotics are functionally unique to *M. tuberculosis* as they target the *M. tuberculosis* cell wall that is structurally distinct from that of other bacteria. Recently, the most active synthesized antitubercular agents have been classified by their chemical structures-amines, amino alcohols, hydrazides, ureas, thioureas, heterocycles, etc. [2,3]. Natural products or their semi-synthetic derivatives also represent an interesting option in the search for new anti-tuberculosis drugs [4,5]. Examples of such semi-synthetic compounds include the well-known antibiotics streptomycin and kanamycin (isolated from *Streptomyces griseus*), and capreomycin (from *S. capreolus*) [4]. Rifampicin is a semi-synthetic drug that was derived from rifamycin, a product of *Amycolatopsis mediterranei* [6]. However, in recent decades only a few new drugs (Bedaquiline, Delamanid, Linezolid, and Clofazimine) have been approved for use in treatment regimens for patients with MDR TB. Although this may seem great, unjustified hopes have been put on the Bedaquiline in particular. After its introduction into clinical practice, it proved to be not very effective, with unwanted side effects such as the QT interval prolongation and its cardiac safety [7,8].

Nitroimidazoles (e.g., delamanid) and nitrofurans are bioreducible drugs whose action is based on the reduction in the nitro group by reductase enzymes. As an alternative to the nitroimidazoles, nitrofuranes (nitrofuranylamides, nitrofuranylpiperazines, nitrofuranylisoxazolines) also demonstrate in vitro and in vivo anti-TB activity [9]. Some new nitrofurans were efficient against actively growing and latent mycobacteria with unique modes of action [10]. Some structure-activity relationship studies showed that the nitro group is essential for the anti-TB activity. Replacement of the furan moiety with another ring leads to a decrease in or lack of activity, while nitroaromatic systems significantly increase the activity against latent or anoxic bacteria [10,11,12].

The main known mechanism of action of nitrofurans as shown for *E. coli*, relies on the activation of the nitrofuran prodrug by nitroreductases leading to oxidative stress due to bactericidal reactive oxygen and nitrogen species [13]. In *E. coli* type I nitroreductases NsfA and NsfB are oxygen-insensitive and catalyze the reduction in the nitro moiety into reactive nitroso and hydroxylamino derivatives. More potential unknown enzymes may exist with less pronounced effect due to low protein expression or low affinity for the nitrofurans [13]. De novo–selected nitrofurantoin-resistant *E*. *coli* strain with wild-type *nfsA* and *nfsB* contained an in-frame deletion in *ribE* that encodes an enzyme in the biosynthesis of flavin mononucleotide, an essential NfsA/NfsB cofactor [14].

The information on the nitrofuran mode of action on mycobacteria and the molecular mechanism of mycobacterial resistance to nitrofurans is limited. *M. tuberculosis* lacks plasmids and horizontal gene transfer and most of its diversity is driven by the chromosomal point mutations or short indels, including the development of drug resistance. Different spontaneous mutations emerge in the *M. tuberculosis* population and may be selected and fixed if they are sufficiently beneficial for bacterial survival, adaptation, and fitness. In this sense, culturing the bacteria on a medium containing an active compound under elevated concentrations may permit the identification of such resistance-associated mutations and gain insight into the mode of action of the compound.

In this study, we describe the synthesis of the new nitrofuranyl amides and investigate their anti-TB activity and possible mechanism of action/resistance through whole-genome sequencing of *M. tuberculosis* spontaneous mutants. We focused on nitrofuranyl amides since they possess strong antitubercular and antibacterial activity. However, especially in the case of antitubercular activity, their mechanism of action is still largely unknown.

A classical design of this kind of mutagenesis study is based on individual resistant clones that are additionally recultured on a drug-containing medium and each individual clone is separately submitted to DNA extraction and whole-genome sequencing. In contrast, in this study, we have purposefully pursued another approach that may be seen as a kind of “metagenomics-like” one. We performed a WGS analysis of the originally grown pooled colonies, rather than single colonies. In this way, we expected to dissect the primary genetic response of mycobacteria to the inhibiting action of the compound.

## 2. Results

### 2.1. Chemistry

A series of six new nitrofuranyl amides was synthesized (Figure 1) by the implementation of a classic methodology for the preparation of amides—namely the reaction of 5-nitrofuran-2-carbonyl chloride (**1**) with different amines (**2**–**7**) in dry dichloromethane (DCM) at basic conditions (ensured by an excess of triethylamine). Intermediates **2**–**7** (except aminoalcohol **3** [15]) are commercial products. All target compounds **8**–**13** were isolated in high purity after column chromatography in moderate to high yields. They were analyzed by using NMR, MS, melting points, and elemental analyses. Detailed synthetic procedures and analytical data are presented in Appendix A.

The choice of the amide moieties as pharmacophore groups in our study is not accidental. Different nitrofuranyl amides are known to be active in vitro against *M. tuberculosis*, but their activity can be significantly influenced by other part of their molecules. Synthesis of **8** was inspired by other active nitrofuranyl anilides [16]. The design of **9** was suggested by our previous studies [15,17] revealing that some fenchone derivatives possess antitubercular activity. Compound **10** combines both nitrofuran and nitroimidazole moieties in one molecule. It is known that bicyclic nitroimidazole PA-824 is a pro-drug with a very complex mechanism of action active against both replicating and hypoxic, non-replicating *Mycobacterium tuberculosis* [18]. The other three compounds (**11**–**13**) in this study contain aryl piperazine moieties, which can contribute significantly to their antitubercular activity [19].

### 2.2. Determination of MIC for Compounds ***8**–**13***

The MIC values of the synthesized compounds were determined for reference strain H37Rv. The compounds were initially tested using both whole-cell microdilution (WCMD) and Resazurin Microtiter Assay (REMA) methods (Table 1). The three most efficient substances with low MIC were further retested using the REMA method in replicated experiments under different ranges of concentrations (Table 2). Replication of experiments was used to eliminate any possible mistakes in the results. To compare compounds **8**–**13** (which possess different molecular weights), all MIC results were presented and commented only in μM. Compounds **12** and **13** showed high efficiency in only one experiment. Repeated testing revealed a very large heterogeneity in the MIC results. This might be due to low solubility or instability of compounds **12** and **13** in the testing media. However, our experiments were oriented only to find an appropriate compound in this series with a low and reproducible MIC value, regardless of the reasons for such heterogeneity. Compound **11** was the only one that showed both low MIC with high reproducibility and concordance of results in different REMA experiments. For this reason, **11** was selected as a model compound for further genetic experiments.

The REMA MIC determination was also performed for a known antibiotic, Isoniazid (MIC 0.062 µg/mL) and confirmed that the condition used to determine MIC was appropriate.

### 2.3. Spontaneous Mutagenesis, Whole Genome Sequencing, and Bioinformatics

The in vitro mutagenesis was performed on *M. tuberculosis* reference strain H37Rv subcultures grown under increasing concentrations of compound **11** (MIC 0.50 µM, Table 2). Whole-genome sequencing (WGS) of the resistant mutants identified mutations in six genes (Table 3).

Strain H37Rv is known to have undergone laboratory evolution and its subcultures are not genomically identical in different laboratories worldwide. Since the first whole-genome sequence of this strain was published in 1987 [20], more H37Rv strains from different laboratories have been sequenced and deposited in GenBank. For this reason, and to avoid false SNP calling, we performed WGS and SNP mapping, not only of the treated with **11** bacterial subcultures, but also of their parental substrain H37Rv used in our laboratory in St. Petersburg.

None of the six mutations were present in the parental strain herein used. These mutations emerged in response to the nitrofuran action (see example in Figure 1). Sufficiently high sequencing depth permitted us to quantitatively and statistically assess the coexistence of the wild type and mutant alleles in the same genome position. In all instances, the mutant reads constituted a minority of all reads, but in all cases, there was a clear increase in the proportion of mutant reads with increasing concentration of the compound, and in some cases, it was a two-fold increase. In four cases, the higher percentage of mutant reads was significant (Table 3).

These six genes are separated by at least 10 kb and it was not possible to assess if the identified mutations were co-segregated on the same chromosome/clone. A simple summation of all percent values gives >100%, i.e., at least some of the mutations co-occurred in the same chromosomes. Since some of the involved genes are related to the oxidative stress response (see below), this is not unexpected as such mutations could be functionally related and would act epistatically.

Notably, five of the six changes are non-synonymous mutations. Analysis using the SIFT (Sorting Intolerant From Tolerant) tool demonstrated that three of them were high-confidence changes that affect protein function (Table 3). Some of the mutations have a very low Point Accepted Mutation 1 (PAM1) values meaning their extremely low probability. These in silico predictions seem to suggest a beneficial effect of such mutations, in case of their emergence and fixation, to the cell survival, counteracting the action of **11**. Given the high importance of at least some of these essential enzymes in wild-type form, the presence of mixed alleles (wild-type and mutant) in the bacterial population suggests these mutations do not appear to affect the in vitro viability of the mutant.

In contrast, a synonymous mutation was found in the *Rv1639c* gene and its increase in response to an elevated concentration of **11** is intriguing but could be associated with random selection driven by a bottleneck effect that may occur as a consequence of in vitro mutant selection [21].

### 2.4. Gene/Protein Enrichment and Network Analysis

We additionally applied gene/protein enrichment and network analysis to the set of the above six mutated genes by making use of the STRING tool. This kind of analysis was shown to be useful in recent studies aimed to decipher the emergence of drug resistance, and to elucidate the potential gene network and signaling pathways in pulmonary TB through gene/protein network analysis [22,23,24].

While no links were detected under the medium confidence score (minimum required interaction score of 0.4), the low confidence score of 0.15 revealed the complex interaction plot shown in Figure 2A that involved four of the six targeted genes. In the network, edges represent protein–protein associations; associations are meant to be specific and meaningful, i.e., proteins jointly contribute to a shared function. The predicted interactions between pairs of these six targeted genes (Figure 2A) based on a different kind of evidence can be briefly summarized as follows. In all three pairs, the significant score was found only for Cooccurrence Across Genomes. However, there was evidence for pairs of *Rv1592–Rv1639c and Rv1592–Rv0224c* of co-expression, reportedly putative interaction, and neighborhood of putative homologs in other organisms (Table 4).

The PPI enrichment *p*-value of 0.464 was non-significant for a network of the six studied genes (Figure 2A). Increasing the size of the network (adding 10 more genes as shown in Figure 2B) resulted in significant enrichment in the larger network with a PPI enrichment *p*-value of 0.00761. Such an enrichment indicates that the proteins may participate or cooperate towards similar metabolic pathways. Although the identified interactions were weak (based on the published body of knowledge assessed by the STRING tool), it may be that future experiments would demonstrate that at least some of the identified putative interactions are true and significant.

## 3. Discussion

The treatment of patients infected with drug-resistant *M. tuberculosis* is still expensive and difficult (often impossible), with significant side effects. The absence of novel types of compounds targeting *M. tuberculosis* encourages researchers to consider well-known antimicrobials and optimize their structures. In particular, nitrofurans are known to be effective against various Gram-positive and Gram-negative bacteria [13,25,26,27]. The 5-nitrofuranyl-piperazine amides are also known derivatives. Aromatic substituents at the second piperazine nitrogen atom can contribute significantly to their anti-TB activity [28,29]. In *M. tuberculosis*, the enzyme Deazaflavin (F420)-Dependent Nitroreductase (Ddn) has the same function as NsfA/NsfB in *E. coli* [30]. The nitro moiety in the nitroimidazole drugs (pretomanid and delamanid) undergoes intrabacterial drug metabolism by the Ddn enzyme. As a result, nitric oxide, a reactive nitrogen species (RNS) is produced, which has bactericidal action [9]. Similar to *E. coli*, the *ribE* gene coding for riboflavin synthase is present in the *M. tuberculosis* genome and could be also involved in the development of resistance to nitrofurans. Nitrofurans are sufficiently well studied with regard to their action on some bacteria, including *E. coli* but even for the latter it was noted that “possible novel nitrofuran resistance mechanism(s), besides well-known *nfsA*/*nfsB* mutations, must receive due attention and resources in order to sustain the utility of this drug class” [13]. In this sense, knowledge of the molecular basis of *M. tuberculosis* resistance to nitrofurans remains limited.

In this study, we sought to gain insight into the mycobacterial resistance mechanism and, respectively, the mode of action of nitrofuranyl amides on *M. tuberculosis*. We used whole-genome sequencing of *M. tuberculosis* spontaneous mutants to achieve a comprehensive assessment of the genetic variation.

Some of the genes herein identified encode proteins involved in response to oxidative stress and some were not previously described as related to resistance to nitrofurans in *M. tuberculosis* or other bacteria. Information on these six genes was retrieved from Mycobrowser and the published literature [31,32,33,34,35,36,37,38,39,40,41,42,43,44,45,46,47,48]; it is summarized (Table 5) and discussed below.

Mutations in *Rv0224c*, *fbiC*, and *iniA* were predicted by SIFT to significantly affect the protein function. Mutations in *fbiC* and *iniA* were concordantly significant as assessed by PAM1 and SIFT scores.

*Rv0224c* is methyltransferase (methylase) in the functional category of Intermediary metabolism and respiration. It is an essential gene (growth defect) for in vitro growth of H37Rv. Previously, its role was investigated in different directions. Firstly, *Rv0224c* is upregulated in response to thiol-specific oxidative stress and protein expression was directly proportional to the transcription [31]. It may be that Rv0224c protein is functionally related to the oxidative stress response. Furthermore, *Rv0224c* codes for a methylase. The primary function of most methylases in prokaryotes is DNA self-recognition via restriction-modification systems that protect the organism against invading DNA. In contrast, “orphan” methylases (without a cognate restriction enzyme) may play an important role in chromosome stability, mismatch repair, replication, influencing gene expression and fitness during hypoxia (see [32] and references therein). The codon 23 of Rv0224c (found mutated in this study) is not located in the methyltransferase domain (http://tbdb.bu.edu/cgi-bin/GeneDetails.html?id=SRv0224c (accessed on 11 September 2022)). On the whole, whether Rv0224c acts as an “orphan” methylase is unknown.

*Rv1173* (*fbiC*) is coding for protein FbiC, which is an essential enzyme for coenzyme F420 production. In turn, nitrofuran prodrug is activated by F420-dependent Ddn nitroreductase. Ddn acts on nitrofuran to release free radicals and other compounds that affect the cell. Low PAM1 value (1) and significant SIFT *p* (0.00) for the identified mutation FbiC Arg774Gly (Table 3) concordantly imply important consequences of this particular amino acid change for cell survival. Previous studies have shown that resistance to the bicyclic nitroimidazooxazine pretomanid is caused by mutations in an F420-dependent bio-activation pathway [33]. The nitro moiety in both nitroimidazole drugs (pretomanid and delamanid) undergoes intrabacterial drug metabolism by the Ddn enzyme. As a result, nitric oxide, a reactive nitrogen species is produced, that has bactericidal action [9]. The cofactor F420 itself is not essential for the survival of *M. tuberculosis* but it protects mycobacteria from antimicrobial compounds by reducing oxidative stress. Inactivation of *fbiC* rendered *M. tuberculosis* hypersensitive to oxidative stress (reviewed in [33]). Structurally, *fbiC* codon 774 is located in the CofH/MqnC C-terminal domain (http://pfam.xfam.org/protein/O50429 (accessed on 11 September 2022)) beyond structural/functional domain termed as “radical SAM family”. Radical SAM proteins are members of a large superfamily of iron-sulfur cluster-containing enzymes that cleave S-Adenosyl methionine (SAM) reductively to generate a 5′-deoxyadenosyl 5′-radical, which allows enzymes to perform a wide variety of unusual chemical reactions [34]. The *fbiC* codon 774 was not described as mutated in a recent survey of mutations related to delamanid resistance [35], nor in a study of spontaneous in vitro-selected delamanid-resistant mutants [36]. However, the latter study described spontaneous mutants in the relatively proximal *fbiC* codons 748 and 792. In view of the above information, this *fbiC774* mutation may be directly mediating resistance to the studied nitrofuranyl amide **11**.

Intriguingly, *ddn* itself did not present any genetic variation in our study. However, a mutation in *fbiC*, a crucial enzyme for Ddn biosynthesis, emerged in response to the nitrofuran treatment. Speculatively, multiple mutants in the same pathway could also lead to reduced fitness. It may be that the physiological fitness of this *fbiC774* mutant was higher than that of hypothetical *ddn* mutants, and one *fbiC* mutation was sufficient to render resistance to nitrofuran.

*Rv0342* (*iniA*) is in the functional category of Cell wall and cell processes, and this gene was shown to be induced, by different mechanisms, by isoniazid and ethambutol. In 2005, Colangelli et al. showed that the *M. tuberculosis iniA* gene is essential for the activity of an efflux pump that confers drug tolerance to both isoniazid and ethambutol [37]. Their results suggest that IniA functions through an MDR-pump like mechanism, although IniA does not appear to directly transport isoniazid from the cell. Wang et al. [38] showed that IniA folds as a bacterial dynamin-like protein (BDLP) with a canonical GTPase domain. IniA does not form detectable nucleotide-dependent dimers in solution. However, lipid tethering indicated nucleotide-independent association of IniA on the membrane. IniA also deforms membranes and exhibits GTP-hydrolyzing dependent membrane fission. These results confirm the membrane remodeling activity of BDLP and suggest that IniA mediates *M. tuberculosis* drug-resistance through fission activity to maintain plasma membrane integrity. Given that isoniazid and ethambutol block cell wall biogenesis, IniA probably contributes to the maintenance of the plasma membrane integrity. Fission of the compromised areas could be used for cell membrane repair [39]. The iniA codon 353 (that was found mutated in our study) is located beyond the GTPase activity domain [38]. However, the low PAM1 value and significant SIFT *p*-value for the mutation Rv0224c Phe23Tyr concordantly suggest that this mutation is biologically meaningful and could contribute to *M. tuberculosis* nitrofuran tolerance through the efflux-related mechanism.

*Rv1592c* encodes a lipolytic enzyme and its expression was up-regulated during isoniazid treatment [40]. This gene is involved in the lipid metabolism of *M. tuberculosis*. Some lipolytic enzymes were shown to promote the survival of bacterium under isoniazid treatment and cell wall lipid remodeling could be the possible reason for drug tolerance. However, the mutation in codon 99 was assessed in silico as nonsignificant and this codon is located outside the lipase domain of the Rv1592c protein (http://pfam.xfam.org/protein/O06598 (accessed on 11 September 2022)).

*Rv1580c* encodes for probable phage protein. It is located in the PhiRv1 prophage that is a possible *M. tuberculosis* complex-specific genomic island [38]. *Rv1580c* is a real protein, that was detected in a rabbit model [42]. Rabbits were experimentally challenged with different Mycobacteria (*M. bovis, M. bovis BCG, M. avium*) and *Rv1580c* was detected 10 weeks post-challenge, suggesting it may be part of a late humoral response.

*Rv1639c* is coding for conserved hypothetical membrane protein. The protein was identified in *M. tuberculosis* H37Rv infected guinea pig lungs at 90 but not 30 days post infection [43]. This is similar to *Rv1580c*, which elicited a late humoral response in the rabbit model. Maloney et al. hypothesized that the two-domain lysyltransferase-lysyl-tRNA synthetase protein negatively regulates *Rv1639c* expression although the exact mechanism remains to be evaluated [44]. In contrast to other non-synonymous mutations detected in this study, a silent mutation was found in the *Rv1639c* gene. According to PFAM database, this codon 404 is located in the putative esterase domain (http://pfam.xfam.org/protein/P94973 (accessed on 11 September 2022)). The reason for the increasing percentage of the mutant allele, in response to an increasing concentration of **11**, is unclear.

It is important to consider that several of these non-synonymous mutations can be unrelated to the resistance phenotype. The fact that one synonymous mutation was detected lends further support to such a notion since the random occurrence of a non-synonymous mutation is generally higher than synonymous ones given the higher number of non-synonymous sites across coding regions [49,50]. However, this and the other five mutations emerged in three independent experiments with the same strain. This is noteworthy and does not seem to be a random occurrence but rather reflects a direct cause-consequence relationship, especially as the mutant alleles increased in a semi-quantitative manner with increasing concentration of the compound.

## 4. Materials and Methods

### 4.1. Chemistry Experimental Procedures

For detailed synthetic procedures and analytical data for synthesized compounds, see Appendix A.

### 4.2. Determination of Minimal Inhibitory Concentration (MIC)

MIC was determined using whole cell microdilution (WCMD) assay and resazurin microtitre plate assay (REMA).

The WCMD was performed as per EUCAST recommendations [51]. Briefly, a bacterial suspension was prepared by adjusting to a 0.5 McFarland standard in sterile water after vortexing 3–4-week cultures grown in Lowenstein–Jensen slopes and subsequently diluting 1:100 in Middlebrook 7H9–10% OADC medium. This final inoculum was used to inoculate U-shaped 96-well sterile plates containing two-fold dilutions of each compound to obtain a final inoculum concentration of 105 CFU/mL. All plates were covered with lids and incubated at 36 ± 1 °C. Each plate included a solvent control, a drug sterility (non-inoculated) control, and at least two drug-free positive controls with a final concentration of 105 CFU/mL and 103 CFU/mL (1:100 positive control). Results were recorded by observation in an inverted mirror as soon as the 1:100 diluted positive control showed visual growth. The MIC was defined as the lowest concentration preventing visual growth.

The REMA (resazurin microtitre plate assay) [52] was performed as described previously [27]. A 3-week *M. tuberculosis* culture was transferred into a dry, sterile tube containing glass beads. The tube was placed on a Vortex shaker for 30–40 s and then 5 mL Middlebrook 7H9 Broth (Becton Dickinson) was introduced. The turbidity of bacterial suspension was adjusted to 1.0 McFarland units (corresponding to approximately 3 × 10^8^ bacteria/mL) and diluted 20-fold with Middlebrook 7H9–10% OADC medium. The same culture medium was used to prepare the 1:100 *M. tuberculosis* (1% population) control. The stock solutions of the compounds in DMSO (10 mg/mL) were diluted with Middlebrook 7H9–10% OADC medium to a concentration of 800 mg/mL. Then, 200 mL of the solution thus obtained was introduced into the 2nd row of a 96-well microtiter plate. This row was used to perform two-fold serial dilutions. Row 10 served as *M. tuberculosis* suspension control, row11—as 1% control (the same culture diluted 10-fold), and row 12—as a blank control for optical density reading (200 mL of the grown medium). The bacterial suspension (100 mL) was introduced into each well, except rows 11 (1% control) and 12 (blank culture medium), to the total volume of 200 mL in each well. The plates were incubated at 35 °C for 7 days. At that point, 0.01% aqueous solution (30 mL) of resazurin (Sigma) was introduced in each well and the incubation continued for 18 h at 35 °C. The fluorescence reading was performed using FLUOstar Optima plate reader. The bacterial viability was determined by comparing the mean values ( ±CI at α = 0.05) of fluorescence in the control wells (row 12, blank and row 11, 1% control) and the wells containing the compound tested. The MIC was determined by the compound concentration, at which the fluorescence reached a plateau or was statistically (*t* criterion) similar to that of the 1% control.

*M. tuberculosis* H37Rv strain used in this assay and for the below genetic analysis originated from the Institute of Hygiene and Epidemiology in Prague in 1976 and was obtained in 2013 from the Federal Scientific Centre for Expertise of Medical Products, Russia, Moscow. The lyophilized strain was inoculated on Loewenstein–Jensen (LJ) growth medium. The 3-week culture was suspended in a physiological solution containing glycerol (15%), transferred into cryotubes, and kept at −80 °C.

The REMA and mutagenesis experiments were performed in the Mycobacteriology reference laboratory at St. Petersburg Institute of Phthisiopulmonology. The laboratory is externally quality assured by the System for External Quality Assessment “Center for External Quality Control of Clinical Laboratory Research” (Moscow, Russia). The REMA method was implemented in the laboratory within the frames of a multicenter European project [53], and was shown to be reproducible in comparison with the reference Bactec MGIT 960 system (Becton Dickinson, Baltimore, MD, USA).

The REMA MIC determination was also performed for a known anti-TB drug (Isoniazid) to affirm that the condition used to determine MIC is appropriate.

### 4.3. Spontaneous Mutant Selection, WGS, and Bioinformatics

In brief, *M. tuberculosis* H37Rv strain was cultured without adding **11** (control culture) and at the elevated concentrations of 11 at 2×, 4×, and 8 × MIC (relative to the MIC determined by REMA). The MIC for strain H37Rv was determined by the REMA method to be 0.50 µM, therefore the concentrations used were 1.00 µM, 2.00 µM, and 4.00 µM.

The bacteria were initially cultured on the LJ solid medium for 3 weeks. The three-week culture was used to prepare a bacterial suspension of three McFarland turbidity standards in saline buffer with Tween. This suspension was further plated on the Middlebrook 7H9 medium (Difco, Becton Dickinson, Le Pont de Claix, France) supplemented with 10% OADC (oleic acid, albumin, dextrose, and catalase [Becton Dickinson, Baltimore, MD, USA]) and containing three different concentrations of **11**, as well as without it (control plate). One ml of suspension (i.e., 10^9^ bacterial cells) was plated per Petri dish.

The Petri dishes were incubated at 37 °C for 4 weeks and DNA was extracted from the pooled colonies from each Petri dish, i.e., single DNA sample per concentration.

*M. tuberculosis* DNA of the parental strain H37Rv and its three subcultures treated with different concentrations of compound 11 was extracted using a CTAB-based method. Whole-genome sequencing was performed at the HiSeq platform (Illumina). DNA libraries were prepared using ultrasound DNA fragmentation and NEBNext Ultra DNA Library Prep Kit for Illumina (New England Biolabs, Ipswich, MA, USA). The mean read length was 110 bp, and the average genome coverage was 132. The mean percentage of the mapped residues was 99.5%. Data for the *M. tuberculosis* sequenced genomes were deposited in the NCBI Sequence Read Archive (project number PRJNA822415, accession numbers: SRR18572456, SRR18572454, SRR18572457, SRR18572455).

The short sequencing reads (fastq files) of the reference sample (laboratory strain H37Rv described above) and its three subcultures grown under different concentrations of 11 were mapped to the annotated reference genome of strain H37Rv (GeneBank accession number NC_00962.3) by PhyReSE online tool (https://bioinf.fz-borstel.de/mchips/phyresse/ (accessed on 11 September 2022)). Furthermore, the resulting lists of SNPs of the control and three treated with 11 samples were compared and the SNPs specific to the samples treated with 11 were identified. The fastq files were also mapped to the reference genome using Geneious 9 program (Biomatters, Auckland, New Zealand) and the above-identified SNPs were additionally verified.

The significance of amino acid substitutions was assessed using PAM1 values calculated by the PhyResSE online tool. The SIFT tool was used to predict whether an amino acid substitution affects protein function based on sequence homology and the physical properties of amino acids (https://sift.bii.a-star.edu.sg/index.html (accessed on 11 September 2022)).

The construction of the protein–protein interaction network was achieved through the use of the Search Tool for the Retrieval of Interacting Genes (STRING) https://string-db.org/cgi/input?sessionId=bo6UyRy8XyiH&input_page_show_search=on (accessed on 11 September 2022), which helped decipher the relationship between these genes.

## 5. Conclusions

This study experimentally demonstrated a complex multifaceted genetic response of *M. tuberculosis* to the action of the nitrofuranyl amide that concerned multiple genes and different pathways. Six genes contained mutations that emerged in bacterial cultures grown under increased concentrations of nitrofuran, furthermore, the increasing concentrations led to a higher proportion of the mutant alleles. The identified genes belong to different gene categories and pathways. Some of the genes were previously reported as being involved in mediating drug resistance or drug tolerance, and counteracting oxidative and nitrosative stress, in particular: *Rv0224c* (oxidative stress response); *fbiC* (F420 biosynthesis pathway that includes nitrofuran activating F420-dependent nitroreductase Ddn); *iniA*, lipase/esterase *Rv1592c* (efflux pump, maintenance of the plasma membrane integrity). The role of other mutated genes (PhiRv1 phage protein *Rv1580c*, and conserved membrane protein *Rv1639c*) is unclear.

The same mutations were detected in different independent experiments, thus confirming a correlation between the action of the compound and possible resistance mechanism. Furthermore, increasing the compound concentration of the compound led to a semi-proportional and significant (in four cases) increase in mutant alleles. Five of six mutations were non-synonymous and some could likely lead to significant changes in protein structure, especially mutations in *iniA* and *fbiC* with concordantly significant PAM1 and SIFT scores.

Further study should focus on experimental analysis of the role of the identified mutations, in particular by the transcriptomics and proteomics methods. Experiments focused on the interaction between nitrofurans and *M. tuberculosis* essential proteins in the presence and absence of a nitroreductase may shed light on the nitrofuran targets.

## Data Availability

Data is contained within the article and Appendix A.

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
