# Peer review of "Molecular Insight into Mycobacterium tuberculosis Resistance to Nitrofuranyl Amides Gained through Metagenomics-like Analysis of Spontaneous Mutants"

_pharmaceuticals, 2022, doi:10.3390/ph15091136_

Round 1

Reviewer 1 Report

In this work, Mokrousov et al. seeks to propose a new set of Nitrofuran amide compounds as potential TB antibiotics and demonstrate the mechanistic bases underlying the emergence of drug resistance against the compounds. Overall, the manuscript is an important step towards identifying new structural scaffold of nitrofuran amide compound and mode-of-action. While the study is interesting and clinically relevant, there are few points that need to be addressed to strengthen the central hypothesis of the study.

1. WGS result after treatment with varying concentrations of compound 11 showed that 6 different genes are potentially involved in gaining drug resistance against the compound. Most of them are involved in defending reactive oxygen species, possibly hinted by E. coli response to nitrofuran amide compounds. However, this experiment is somewhat incomplete. Currently, accumulating experimental evidence has shown that mycobactericidal antibiotics kill M. tuberculosis by accumulating ROS and RNI thus, mutations on genes involved in the 6 genes may also confer M. tuberculosis resistant to many anti-TB antibiotics. This speculation is also partly supported by Fig 2A, in which four out of 6 genes are interconnected as a defense mechanism against ROS and RNI. Thus, authors should include experiments using the mutants to test if they are also resistant to other antibiotic treatment or other nitrofuran amide compounds.

Authors need to include the putative discussion why Ddn doesn’t belong to the list of genes that are involved in drug resistance to compound 11.

2. M. tuberculosis has a complex and chemically resistant cell wall. Therefore, anti-tubercular drugs are specific and do not act on other pathogenic bacteria, and vice versa many available antibiotics do not (with a few exceptions) act on mycobacteria: Many TB antibiotics are functionally unique to Mtb as they target Mtb cell wall that is structurally distinct from that of other bacteria. 

3. The main known mechanism of action of nitrofurans as shown for E. coli, relies on the activation of the nitrofuran prodrug by nitroreductases leading to oxidative stress due to bactericidal ROS and RNI: needs references.

4. WGS identified 6 mutations in 6 genes. Do these cause functions of these proteins or are these known to alter their catalytic activities ?

5. Table 1: MIC determination needs to include the test using known TB antibiotics to affirm that the condition used to determine MIC is appropriate.

6. In Material and Method section 4.2, please describe WCMD and REMA methods briefly.

7. Tables 1 and 2 showed that REMA method is not very reproducible. This reviewer doesn’t understand why authors use REMA to confirm the Table 1 result. CFU mediated MIC determination would be more prevalently accepted in TB research area.

8. Result 2.2. Authors should show the solubility and stability in the media using LCMS analysis. Because Compounds 12 and 13 showed much better antimycobacterial effects as compared to that of compound 11 although the MIC values deduced by WCMC and REMA methods.

9. In section 2.2, authors said that compound 11 is structurally stable. This reviewer can’t find any experimental evidence.

Reviewer 2 Report

It is good written study to understand the impact of nitrofuranyl amides on H37Rv. It is very interesting to see same mutation in the independent exposures and this indicates the importance of mutations. I have following observations that might be helpful for improving the manuscript.

1. As there is no high confidence in protein interaction, is it worth reporting?

2. Is there any mutation that is not common in all the independent exposures? if yes, does this indicate anything from literature

3. Understress, the genomes of bacteria tend to recombinate around repeat elements, is there any observation in H37Rv under stress in your experiment

Round 2

Reviewer 1 Report

All concerns are addressed.